# BE4max and AncBE4max Are Efficient in Germline Conversion of C:G to T:A Base Pairs in Zebrafish

**DOI:** 10.3390/cells9071690

**Published:** 2020-07-14

**Authors:** Blake Carrington, Rachel N. Weinstein, Raman Sood

**Affiliations:** Zebrafish Core, Translational and Functional Genomics Branch, National Human Genome Research Institute, National Institutes of Health, Bethesda, MD 20892, USA; carringb@mail.nih.gov (B.C.); rachel.weinstein@nih.gov (R.N.W.)

**Keywords:** zebrafish, base editing, genome editing, BE4max, AncBE4max

## Abstract

The ease of use and robustness of genome editing by CRISPR/Cas9 has led to successful use of gene knockout zebrafish for disease modeling. However, it still remains a challenge to precisely edit the zebrafish genome to create single-nucleotide substitutions, which account for ~60% of human disease-causing mutations. Recently developed base editing nucleases provide an excellent alternate to CRISPR/Cas9-mediated homology dependent repair for generation of zebrafish with point mutations. A new set of cytosine base editors, termed BE4max and AncBE4max, demonstrated improved base editing efficiency in mammalian cells but have not been evaluated in zebrafish. Therefore, we undertook this study to evaluate their efficiency in converting C:G to T:A base pairs in zebrafish by somatic and germline analysis using highly active sgRNAs to *twist* and *ntl* genes. Our data demonstrated that these improved BE4max set of plasmids provide desired base substitutions at similar efficiency and without any indels compared to the previously reported BE3 and Target-AID plasmids in zebrafish. Our data also showed that AncBE4max produces fewer incorrect and bystander edits, suggesting that it can be further improved by codon optimization of its components for use in zebrafish.

## 1. Introduction

Genetically engineered animal models play a vital role in the study of genes mutated in human diseases. In general, when mutations in a gene are identified in human patients, animal models are developed to prove disease causation (display of similar phenotype), understand disease mechanism and identify potential treatments. Zebrafish offer several advantages as a vertebrate animal model, such as external fertilization, optically transparent embryos, rapid embryonic development, large clutch size, evolutionarily conserved biological pathways and suitability for high-throughput mutagenesis and chemical screening [1,2,3,4]. Targeted genome editing by CRISPR/Cas9 technology provides an inexpensive, easy to use, and versatile method to generate gene knockout zebrafish models [3,5]. Several laboratories including ours have established high-throughput methods to generate zebrafish with knockout of single or multiple genes using CRISPR/Cas9 [6,7,8,9,10]. Briefly, small guide RNA (sgRNA) designed to target a desired site (exon or functional domain) in the gene of interest are co-injected with Cas9 mRNA or protein into one-cell-stage zebrafish embryos. Cas9 is guided to the desired site in the genome by recognition of the protospacer adjacent motif (PAM) which is one of the requirements of sgRNA design. Upon binding to the target site, Cas9 induces a double-strand break (DSB) which activates DNA repair pathways of non-homologous end joining (NHEJ) and homology directed repair (HDR). Generation of gene knockout mutants relies on the highly efficient and error-prone NHEJ repair that leads to insertion or deletion (indels) of a few to several nucleotides. Out-of-frame indels leading to frameshift and premature truncation of the protein can be selected to establish knockout fish lines. 

More than half of disease-causing mutations in human genetic diseases are caused by single-nucleotide changes that lead to missense, nonsense, stop-loss, or splicing mutations [11]. Except for the nonsense mutations, their mechanisms of action are often predicted using computational methods and require validation in animal models [12,13,14]. Missense mutations can either cause loss or gain of function (dosage effect) or alter the function of the mutant protein (new function). Similarly, splicing mutations can lead to exon inclusion, intron retention or activation of cryptic splice sites [15]. Thus, knockout zebrafish models may not recapitulate the phenotypes caused by such point mutations. Instead, methods to generate zebrafish models with exact nucleotide change as seen in the patients are required to study these mutations. The commonly used method to generate zebrafish with single-nucleotide substitutions takes advantage of the CRISPR/Cas9 based knockin using HDR [16,17,18,19]. A repair template with the desired nucleotide change and homology arms is synthesized as a single-stranded or double-stranded donor DNA and injected along with the Cas9 (protein or mRNA) and sgRNA. However, limitations such as possibility to design an active sgRNA close to the intended site of nucleotide change, unintended indels due to the simultaneous activation of NHEJ, and low efficiency of knockin has made it difficult to apply HDR-based approaches for single-nucleotide substitutions in a high-throughput manner in zebrafish [20,21]. Furthermore, the design of donor DNA is not straightforward and several versions with homology arms of different lengths need to be tested for each site [10,22]. 

Recently developed programmable base editing technology that combines the targeting specificity of Cas9 with a cytidine or adenine deaminase provides an alternate to HDR for generation of zebrafish with C:G to T:A or A:T to G:C substitutions [3,23,24,25,26,27,28]. In a typical base editor, nCas9 is fused to cytidine or adenine deaminase with small linkers [29]. The purpose of using a nickase version of Cas9 is to induce nicking of the unedited strand without causing a DSB, thus preventing activation of NHEJ and unintended indels [11,30]. Cas9 can be replaced with its natural or engineered variants or Cpf1 to improve the specificity and targeting scope of base editors [29]. Cytidine deaminases induce the conversion of cytosine (C) in the targeting window (usually five nucleotides in position −13 to −17 from PAM) to uracil (U), which pairs with adenine (A), thus causing a C to T (or G to A on the opposite strand) change in the target DNA [3,29,30] (Figure 1A). Adenine deaminases induce the conversion of A in the targeting window to inosine (I), which pairs with C, thus causing an A to G (or T to C on the opposite strand) change in the target DNA [3,29,31]. In cytosine base editors, uracil DNA glycosylase (UGI) is used either by co-expression or as a fused component to inhibit the removal of U by the base excision repair (BER) pathway [29,30,32,33]. 

Komor and colleagues [30] were the first to develop a set of cytosine base editors (CBE) termed BE1 to BE3 by using APOBEC-1 from rat as the cytidine deaminase. In the same year, Targeted AID-mediated Mutagenesis (TAM) and Target-AID, that used cytidine deaminase AID from human or sea lamprey, respectively, were reported [32,33]. Subsequent studies focused on modifications, such as codon optimization or substitution of Cas9 with its variants, such as HF-Cas9, VQR-Cas9, or with Cpf1 to improve the specificity and targeting scope by use of different PAM sequences [23,24,34,35,36,37]. Although editing of desired C or G nucleotides to T or A was demonstrated using these CBEs in a variety of cells and organisms including zebrafish, their overall efficiency was low and they also caused unintended nucleotide changes and indels at the target site [23,24,28,37,38,39]. Since then, notable improvements have been made to BE3, leading to the development of an improved and optimized set of cytosine base editors, termed BE4, BE4max and AncBE4max [40,41]. These modifications included an addition of a second copy of UGI to the C terminus of BE3 for improved inhibition of BER pathway (BE4), addition of bipartite nuclear localization signals at both N and C-termini (BE4max), and substitution of rAPOBEC-1 with APOBEC optimized by ancestral reconstruction (AncBE4max) (Figure 1B). Both BE4max and AncBE4max plasmids can also have a P2A linked GFP that acts as a readout of protein synthesis (Figure 1B).

To date, only proof-of-principle base editing experiments using the earlier versions of CBEs, Target-AID, BE3, and modified BE3 with different Cas9 variants, have been reported in zebrafish [23,24,25,28]. Although targeted base editing was observed as expected in these studies, its efficiency was low and unwanted indels were observed. As the process of screening for base editing in zebrafish requires extensive cloning and sequencing, it is highly desirable to have efficient base editors to reduce the labor and costs associated with developing humanized fish models. With the goal of identifying efficient cytosine base editors in zebrafish, we undertook this study to evaluate the performance of BE4max and AncBE4max with and without GFP (Figure 1B). Here, we present our data on their base editing efficiencies at two target loci by somatic and germline screening.

## 2. Materials and Methods

### 2.1. Zebrafish Husbandry

All experiments were performed using the wildtype TAB5 zebrafish strain on the National Human Genome Research Institute Animal Care and Use Committee (ACUC) approved protocol G-05-5. Zebrafish husbandry and breeding were performed as previously described in the Zebrafish book [42]. 

### 2.2. sgRNA Selection, sgRNA and Cas9 mRNA Synthesis

sgRNA targets were selected based on previously reported successful base editing with BE3 in zebrafish [24]. Their sequences are shown in Table 1 with target Cs marked in red color and PAM in blue color. sgRNAs and Cas9 mRNA were synthesized using previously described protocols [43]. 

### 2.3. Base Editing Constructs and RNA Synthesis

Plasmids for the following four base editing constructs: pCMV_BE4max (Addgene plasmid # 112093), pCMV_BE4max_P2A_GFP (Addgene plasmid # 112099), pCMV_AncBE4max (Addgene plasmid # 112094), and pCMV_AncBE4max_P2A_GFP (Addgene plasmid # 112100) [40] were kind gifts from Dr. David Liu. Plasmid DNA was purified from an overnight culture of TOP10 cells (Thermo Fisher Scientific, Waltham, MA, USA) using a miniprep kit (Qiagen, Germantown, MD, USA). All four plasmid DNAs were linearized with SapI (New England BioLabs, Ipswich, MA, USA) and mRNA was synthesized with the T7 mMessage mMachine kit (Thermo Fisher Scientific, Waltham, MA, USA) according to the manufacturer’s instructions with the following modifications: addition of 1 µL GTP, an incubation time of 3 h and LiCl precipitation.

### 2.4. Zebrafish Injections

For the evaluation of sgRNA activity, embryos were co-injected into the yolk at the one-cell stage with 50 pg sgRNA and 300 pg spCas9 mRNA. For base editing experiments, embryos were co-injected into the yolk at the one-cell stage with 50 pg sgRNA and 600 pg of one of the base editor mRNAs. All injected embryos were evaluated for toxicity at 24 h post fertilization (hpf).

### 2.5. DNA Extraction, Flourescent PCR for CRISPR-STAT and Sanger Sequencing

DNA extraction, fluorescent PCR for CRISPR-STAT and Sanger sequencing were performed as previously described [43,44]. Briefly, DNA was extracted from individual or pooled embryos using the Extract-N-Amp kit (Sigma-Aldrich, St. Louis, MO, USA) at one quarter of the recommended volumes. CRISPR-STAT was performed with 8 individual embryos collected from each injection group at 1 day post fertilization (dpf) and 8 individual uninjected embryos with the following gene specific primers: *twist2* forward (AGGAGCTGGACAGACAGCAG), *twist2* reverse (TCAATGTACCTGGATGCGAG), or *ntl* forward (CAACAGAAGTGACCACAAGG) and *ntl* reverse (TTGCTCTTACTGGTGGTAGTGC). M13F and PIGtail adapters were added to the forward and reverse primers, respectively, and a third FAM-M13F primer was used for fluorescent labeling of the PCR product. PCR conditions were as follows: 12 min at 95 °C; 40 cycles 94 °C for 30 s, 57 °C for 30 s, 72 °C for 30 s; 72 °C for 10 min. Fluorescent PCR products were run on a 3730 Genetic Analyzer and CRISPR-STAT analysis was performed by comparing the number of peaks in injected embryos compared to their uninjected controls. PCR amplification for sequencing was performed with the same primers and conditions except for the substitution of *ntl* forward primer with a different M13F-tailed *ntl* sequencing forward primer (CAACAGAAGTGACCACAAGG) in order to obtain a high-quality sequence at the target site. PCR products were treated with ExoSAP-IT (Thermo Fisher Scientific, Waltham, MA, USA) followed by Sanger sequencing with M13F primer using the BigDye Terminator v3.1 kit (Thermo Fisher Scientific, Waltham, MA, USA). 

### 2.6. Somatic Base Editing Analysis

From each injection group, 24–48 embryos were collected individually in a 96-well PCR plate at 48 hpf and processed for DNA extraction, PCR and Sanger sequencing using the M13F primer as described above. Sequence chromatograms were analyzed in Sequencher (Gene Codes, Ann Arbor, MI, USA) to identify embryos showing secondary peaks in the target window. We analyzed all sequence chromatograms that showed secondary peaks in the 5 bp target window using the online EditR tool (http://baseeditr.com) by following the “How to use EditR” instructions. Quad plots were used to validate secondary peaks and editing was considered positive when the trace nucleotide(s) had significance above the *p*-value cutoff of 0.01. 

### 2.7. Screening for Germline Transmission of Edited Nucleotides 

Injected embryos were grown to adulthood and screened by pairwise crosses with WT fish. At 1 dpf, embryos from the progeny were collected as pools of 4 embryos/well in a 96-well PCR plate. A minimum of 10 pools/founder fish were screened and in some cases the founder fish were crossed twice due to small clutch size. DNA extraction, PCR and Sanger sequencing using the M13F primer were performed as described above. Sequence chromatograms were analyzed in Sequencher to identify embryos showing secondary peaks in the target window. Germline transmission was then confirmed by re-breeding of all positive founders and sequencing of individual embryos (minimum of 48 embryos/founder fish).

## 3. Results

### 3.1. Experimental Design 

The objective of our study was to evaluate improved cytosine base editors, BE4max and AncBE4max, with and without GFP (Figure 1B) for their effectiveness in precision base editing in the zebrafish. Therefore, in order to minimize variables that may have an effect on the outcome of editing, we selected sgRNAs that have previously been shown to induce C → T or G → A conversion in the zebrafish using BE3 [24]. Two of these sgRNAs were designed to target the *twist* locus and one was designed to target the *ntl* locus (Table 1). To make sure that these sgRNAs would efficiently bind their target DNA in our zebrafish strain, we first tested them by CRISPR-STAT to assess their ability to generate indels at the target site with spCas9. For each sgRNA we evaluated eight injected embryos for somatic activity. Our data showed that *twist2*-T1 (6/8 embryos) and *ntl* (7/8 embryos) sgRNAs caused indels detected as multiple peaks, whereas *twist2*-T2 (0/8 embryos) failed to cause indels at detectable levels in our strain of fish (Figure 2A). Based on these data, *twist2*-T1 (hereafter referred to as *twist2*) and *ntl* sgRNAs were used for subsequent base editing experiments. Both target sites contain two Cs in the targeting window that can be converted to T with CBEs (Table 1).

Our strategy involved screening for somatic and germline base editing activity of BE4max and AncBE4max with and without GFP at *twist2* and *ntl* loci in the zebrafish as depicted in Figure 2B. Embryos were injected with sgRNA and base editor mRNA and then evaluated at 48 hpf for somatic base editing using Sanger sequencing and EditR analysis of chromatograms. If base editing was observed at the somatic level, we pursued those base editors to test for germline transmission of edited nucleotides. This was achieved by injecting ~100 additional one-cell-stage embryos and growing them to adulthood. A subset of these injected embryos was re-screened for somatic activity to rule out any technical issues during repeat of injections. When embryos reached the adult stage, they were crossed with wildtype fish and progeny were screened by pooling four embryos/well. This strategy was designed to screen maximum number of embryos/founder fish and still be able to screen multiple founders/plate while reducing labor and cost associated with DNA extraction, PCR and sequencing. All founder fish that showed evidence of base editing in pooled embryos were re-screened by sequencing of an additional 48 individual embryos to confirm that the F1 progeny were heterozygous for the edited allele (Figure 2B).

### 3.2. Somatic Base Editing

Previous studies have demonstrated a positive correlation between somatic and germline mutations using CRISPR/Cas9 in zebrafish [44,45]. Therefore, we performed an evaluation of each plasmid at both loci for somatic base editing first, as it is quicker than the germline transmission evaluation. We chose to evaluate the GFP versions of BE4max and AncBE4max plasmids first, so that injected embryos could be identified by the presence of GFP. However, no GFP was detected in injected embryos at 8, 24 or 48 hpf and therefore BE4max-P2A-GFP and AncBE4max-P2A-GFP plasmids were not pursued further. The lack of GFP expression could be due to the inability to synthesize full-length mRNA for these plasmids due to their large size (~7 kb). Somatic base editing was observed in 4% to 11% of embryos injected with BE4max and AncBE4max mRNAs without the linked GFP (Table 2, Figure 3). In all cases, base pair changes occurred at the expected nucleotides and the substituted nucleotides were as expected (C → T or G → A). These data suggest that BE4max and AncBE4max are capable of base editing at the somatic level in zebrafish. 

### 3.3. Germline Transmission of Edited Nucleotides

The generation of stable zebrafish lines with targeted nucleotide substitutions is essential to study specific alleles. Since germline transmission rates can greatly vary [7], we implemented a 2-step strategy to reduce labor and costs associated with screening. First, we identified potentially positive founders by screening a minimum of 40 embryos from each founder as pools of 4 embryos/well for the presence of an edited allele (Figure 4). We manually inspected the sequence chromatograms at target nucleotides for the presence of secondary peaks as they may be a small fraction of the main peak if only one of the eight alleles in the pool has been edited. This pooling method allowed detection of germline transmission at both targets by both plasmids and allowed us to identify at least one germline transmitting founder fish by screening four to seven fish per condition (Table 3). We then confirmed base editing by detection of embryos heterozygous for the edited nucleotide by sequencing of a minimum of 48 individual embryos from each positive founder fish (Figure 4). Interestingly, these sequence data revealed a combination of correct and incorrect substitutions at the targeted nucleotides as well as off-target bystander editing outside of the target window in progeny of three out of the five founder fish (Table 3, Figure 4). BE4max was able to correctly edit one of the two targeted nucleotides (C_4_) at *twist2* in two independent founders in 9% and 3% embryos, respectively (Table 3, Figure 4(Bii,iv). However, a majority of the progeny from founder 2 (22%) inherited an incorrectly edited allele (C_4_ → G_4_) (Figure 4(Bv)). In the embryos inheriting correctly edited allele (C_4_ → T_4_) from founder 2, off-target editing (C_11_ → T_11_) was observed outside the target window (Figure 4(Biv). In contrast to BE4max, AncBE4max precisely edited both nucleotides (C_4_C_5_) simultaneously in the target window of *twist2* (Figure 4(Cii)) in 9% of F1 embryos. Although 4% of F1 embryos inherited incorrect or off-target nucleotide changes, none of them were observed in the correctly edited embryos (Appendix A). In the progeny of founder from BE4max at *ntl* locus, we observed a combination of correct (G_5_ → A_5_) and incorrect (G_6_ → T_6_) editing at the two Gs in the target window (Figure 4(Eii)) in 5% of embryos. In contrast, AncBE4max converted only one of the two Gs to A (G_5_ → A_5_) (Figure 4(Fii)) in 7% of embryos and no off-target edits were observed. We did not evaluate our fish for genome-wide off-target effects, as we can use out-crossing and genotype-phenotype correlations to analyze the mutant fish. Overall, our data showed that while off-target and incorrect editing can occur, both BE4max and AncBE4max are successful in efficiently editing the germline in zebrafish with no observable indels. Our data also suggest that the AncBE4max with ancestral cytidine deaminase outperforms BE4max in overall correct editing as well as having less off-target effects.

## 4. Discussion

Zebrafish have emerged as an excellent intermediate model between invertebrate (worm and fruit fly) and mammalian (mouse) models to study human genetic diseases [46]. Knockout zebrafish mutants have become easy to generate due to advances in genome editing technologies, such as Zinc finger nucleases, transcription activator-like effector nucleases and CRISPR/Cas9 [27,47]. Although knockout mutants make excellent models to study loss-of-function mutations, they fail to recapitulate the phenotypes of pathogenic point mutations. The evolution of base editors has provided a new method to generate animal models with targeted single base pair changes to mimic human mutations in a variety of model systems. Here, we showed that CBEs, BE4max and AncBE4max, can be used to efficiently generate stable zebrafish lines with desired point mutations. The percentage of germline transmitting founders in our study (25–28%) is comparable to that reported with BE3-zCas9 (28–37%) at the same two loci [24]. We have also presented a streamlined method of screening for somatic and germline base editing while reducing labor and reagent costs. We recommend using CRISPT-STAT to ensure that an active sgRNA is selected for base editing experiments. Overall, our data showed that AncBE4max performed better than BE4max, with germline transmission rates of 7.9% (14/177) vs. 4.7% (9/193), respectively (Table 3), which is consistent with the findings in human cell lines [40]. 

As shown in Table 1, we expected editing at two adjacent nucleotides in both *twist2* (C_4_C_5_) and *ntl* (C_5_C_6_). However, with the exception of *twist2*/AncBE4max, correct editing was observed at only one of the two target nucleotides (C_4_C_5_ → T_4_C_5_ in *twist2*/BE4max and C_5_C_6_ → C_5_T_6_ in *ntl with* BE4max and AncBE4max). Komor and colleagues [30,41] showed that editing is context dependent and follows the order TC > CC > AC > GC with the second C being the target nucleotide. Thus, for *twist2*, C_4_ (context TC) would be favored over C_5_ (context CC) and matched with our data. For *ntl*, the context should have favored editing of C_6_ (context CC) over C_5_ (context AC), but opposite was observed. Thus, we speculate that factors other than context of the adjacent nucleotide play a role in correct editing of a target nucleotide. 

Precision and efficiency of editing are important for widespread adoption of a base editor to generate zebrafish models of human diseases. Successful base editing has been demonstrated in zebrafish using earlier versions of CBEs, such as BE3, BE3-zCas9, BE3-VQRCas9, Target-AID [23,24,28]. However, these studies also reported a high prevalence of indels in the edited embryos. We did not observe any indels with BE4max and AncBE4max at either of the two targets. In addition to indels, incorrect conversion of the target nucleotide (e.g., C →A or G), bystander editing (nucleotide conversions elsewhere in the amplicon), and genome-wide off-target edits are reported as common side effects of CBEs [24,28,48,49,50,51,52]. We did not perform an evaluation of genome-wide off-target edits. However, we observed incorrect and bystander edits within the amplicon with BE4max and AncBE4max at both target loci. Incorrect conversion (C_4_ → G_4_ in *twist2* and G_6_ → T_6_ in *ntl*) was observed more frequently with BE4max as compared to AncBE4max. Interestingly, all embryos with editing at *ntl* by BE4max inherited a combination of correct (G_5_ → A_5_) and incorrect conversions (G_6_ → T_6_), thus converting GG to TA (or CC to TA on the reverse strand) in the target window. Bystander edits were observed either in combination with the correct edits at the target site, e.g., *twist2*/BE4max (Figure 4B) or independent of a correctly edited allele, e.g., *twist2*/AncBE4max (Appendix A). Thus, further optimization is required to minimize the incorrect and bystander edits by BE4max and AncBE4max.

Previous studies have shown that optimization of genome editing nucleases for their specific use in zebrafish using codon optimization has been effective in improving their efficiencies [24,26,53]. Zebrafish codon-optimized versions of adenine base editors ABE7.10 (zABE7.10) and ABEmax (zABE7.10max), where both adenosine deaminase and nCas9 were replaced with their zebrafish codon-optimized counterparts (zTadA, zTadA* and znCas9), showed significantly increased editing at several loci as compared with the original ABE7.10 [26]. Thus, we can speculate that similar optimization of BE4max and AncBE4max base editors would increase their efficiencies in the zebrafish. Such optimized base editors can be easily modified by swapping nCas9 with its variant versions or Cpf1 to allow use of different PAM sites in cases where an NGG PAM is not available at the required distance from the nucleotide to be edited. Efficient base editors would facilitate the creation of clean knockout alleles with stop codons instead of CRISPR/Cas9-induced indels [54,55]. It has been shown that occasionally CRISPR/Cas9-induced indels lead to activation of cryptic splice sites and mask the knockout phenotypes by producing partially functional protein [56,57]. Similarly, efficient base editors can be used to modify splice acceptor sites to generate zebrafish with skipped exons to study functions of specific isoforms [58].

In conclusion, our study showed that BE4max and AncBE4max can be used to generate heritable point mutations in zebrafish without unintended indels. Overall, AncBE4max showed higher editing efficiency and less incorrect base pair conversions or bystander edits compared to BE4max. Therefore, these CBEs make a valuable addition to the zebrafish toolbox for generation of desired point mutations. 

## Figures and Tables

**Figure 1 cells-09-01690-f001:**
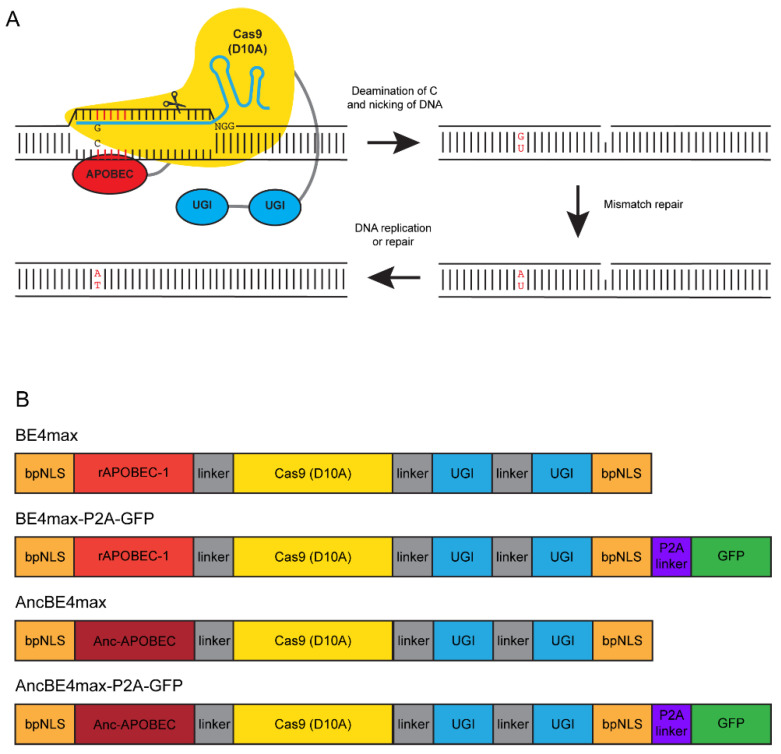
An overview of C → T conversion by BE4 and cytosine base editors used in this study. (**A**) A schematic of base editing depicted as conversion of C to T by BE4. The base editor is localized to the target DNA with a sgRNA (blue line and hairpins). Cytidine deaminase APOBEC-1 (red) performs deamination of the C to U and the Cas9n (yellow) nicks the unedited strand. Two copies of UGI (blue) are used to prevent removal of U and subsequent endogenous DNA mismatch repair and replication leads to C → T conversion as depicted. (**B**) A linear representation listing various components of BE4max or AncBE4max with and without GFP set of plasmids used in this study (based on [40]). bpNLS stands for bipartite nuclear localization signal.

**Figure 2 cells-09-01690-f002:**
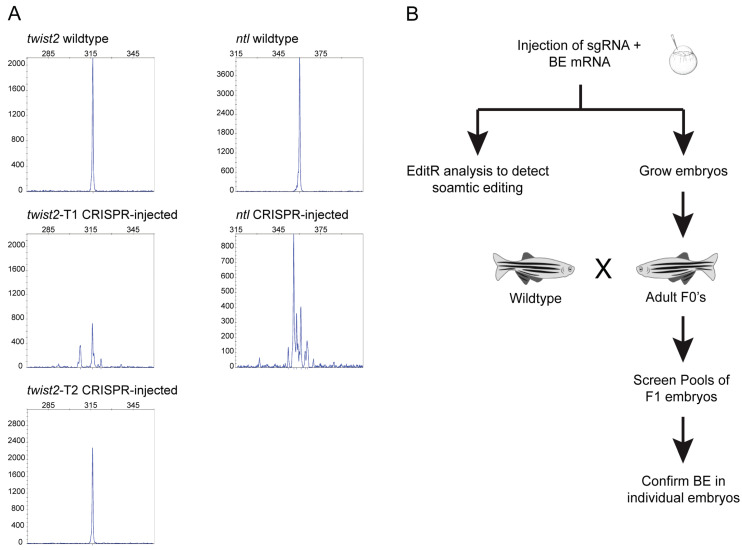
Selection of targets and strategy for assessment of base editing efficiency. (**A**) Evaluation of sgRNA targets by CRISPR-STAT. Peak profiles from representative wildtype (uninjected) or sgRNA-injected embryos are shown. *twist2*-T1 and *ntl* sgRNAs showed multiple peaks indicating DSB/NHEJ repair, whereas *twist2*-T2 showed no DSB/NHEJ activity. (**B**) Schematic of workflow for screening zebrafish at the somatic and germline levels. Injected embryos were screened using sequencing and EditR analysis for somatic base editing and additional progeny were grown to adulthood. Adult fish were then screened for germline transmission by outcrossing with wildtype fish and screening by sequencing pools of four embryos followed by confirmation of base editing in individual embryos.

**Figure 3 cells-09-01690-f003:**
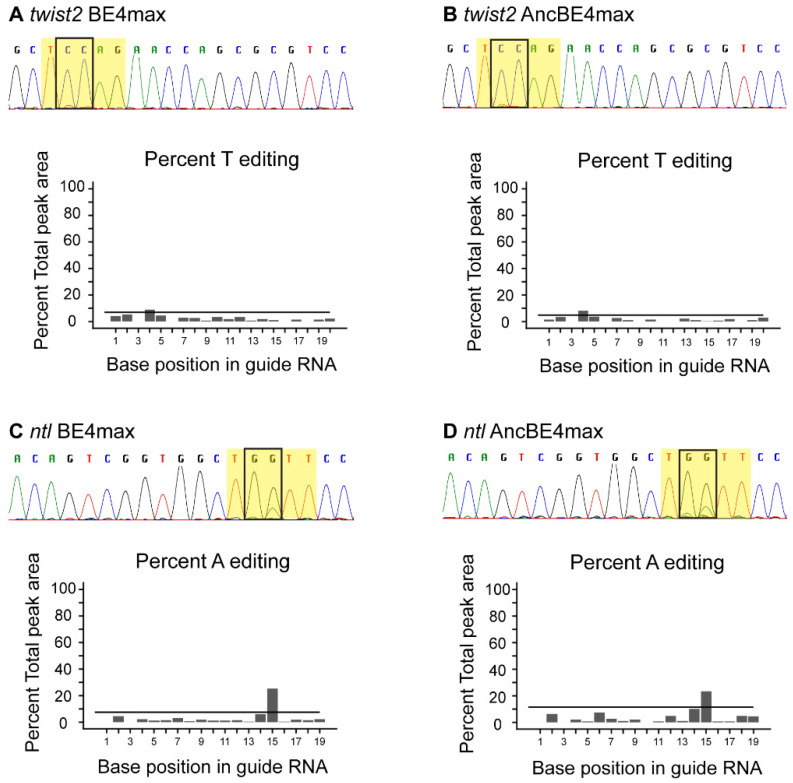
Somatic base editing with BE4max and AncBE4max. (**A**,**B**) somatic base editing at *twist2* locus with BE4max (**A**) and AncBE4max (**B**). (**C**,**D**) somatic base editing at *ntl* locus with BE4max (**C**) and AncBE4max (**D**). In all panels, partial sequence chromatograms of representative embryos are shown with the target window highlighted in yellow and target nucleotides boxed. EditR Quad plots for the expected edited allele are shown below each chromatogram. These plots show the percent editing (Y-axis) at each nucleotide by base position (X-axis), with a horizontal line representing the *p*-value cutoff of 0.01. Any nucleotide above this cutoff is considered a real secondary peak. Expected base pair changes are C → T for *twist2* or G → A for *ntl* (*ntl* sgRNA is on the reverse strand and orientation shown here is to depict the open reading frame).

**Figure 4 cells-09-01690-f004:**
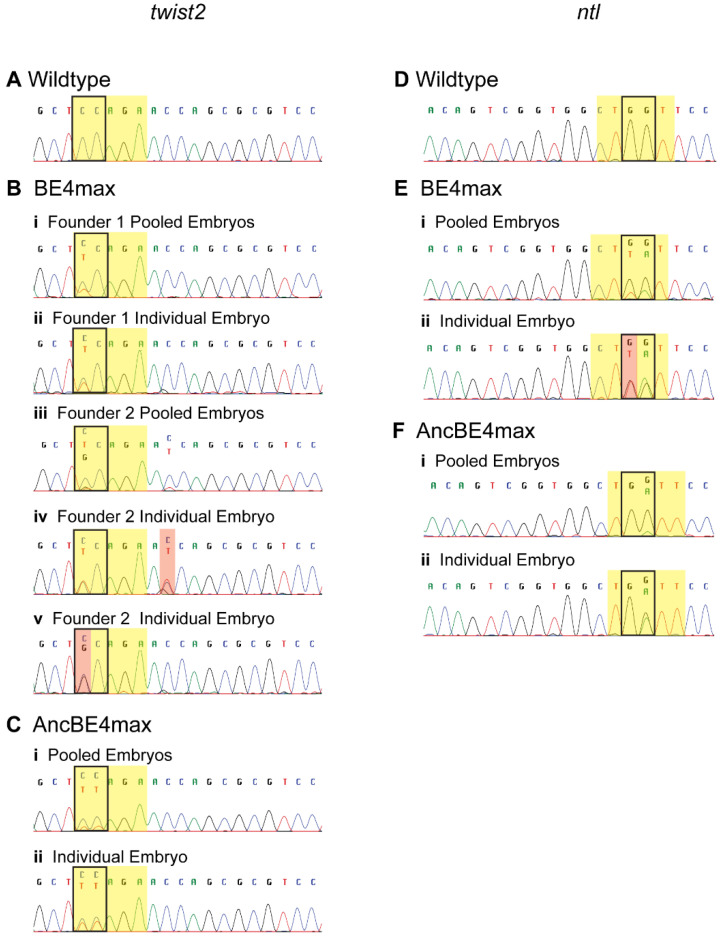
Germline base editing in pooled or individual embryos. Representative chromatograms of a wildtype embryo (**A**,**D**), pooled or individual F1 embryos at the *twist2* (**B**,**C**) and *ntl* (**E**,**F**) loci are shown with the target window highlighted in yellow and target nucleotides boxed. Correct base editing for *twist2* (C → T) and *ntl* (G → A) is observed as a trace peak in pooled embryos or as heterozygous peaks in individual embryos (*ntl* sgRNA is on the reverse strand and orientation shown is to depict the open reading frame). Incorrect bp changes and bystander edits are highlighted in red.

**Table 1 cells-09-01690-t001:** Sequence of sgRNA targets with target Cs marked in red and PAM sites in blue.

Gene-Target	Sequence (5′-3′)
*twist*2-T1	GCTC_4_C_5_AGAACCAGCGCGTCC TGG
*twist*2-T2	GCCGC_5_TC_7_GCGTACGTTCGCC AGG
*ntl (tbxta)* ^1^	GGAAC_5_C_6_AGCCACCGACTGT TGG

^1^*ntl* sgRNA is on the reverse strand and therefore expected nucleotide changes are G → A in its coding sequence.

**Table 2 cells-09-01690-t002:** Details of screening for somatic base editing.

Target	Base Editor	Embryos Screened	Embryos with Nucleotide Changes in Target Window	
			Correct	Incorrect
*twist2*	BE4max	47	2 (4%)	None
*ntl*		47	2 (4%)	None
*twist2*	AncBE4max	47	5 (11%)	None
*ntl*		47	3 (6%)	None

**Table 3 cells-09-01690-t003:** Details of screening for germline transmission.

Target	Base Editor	F0 Fish Screened ^1^	F0 Fish with Base Editing	Individual F1 Embryos	
				Correct Editing	Incorrect Editing
*twist2*	BE4max	7	2	3/34 (9%)	None
				1/37 (3%) ^2^	8 (22%)
	AncBE4max	4	1	7/82 (9%)	3 (4%)
*ntl*	BE4max	6	1	5/122 (4%) ^3^	5 (4%) ^3^
	AncBE4max	4	1	7/95 (7%)	None

^1^ Minimum of 10 pools of embryos screened. ^2^ Correct edit C_4_ → T_4_ in edit window with an additional C_11_ → T_11_ edit outside edit window. ^3^ Correct edit G_5_ → A_5_ and incorrect edit G_6_ → T_6_ in edit window.

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
