# Peer review of "BE4max and AncBE4max Are Efficient in Germline Conversion of C:G to T:A Base Pairs in Zebrafish"

_cells, 2020, doi:10.3390/cells9071690_

Round 1

Reviewer 1 Report

The authors applied a new version of the BE4max or AncBE4max base editor
to zebrafish to facilitate the creation of disease models.
This is important because it extends the genome editor toolbox
to make genome editing more precise in zebrafish system.
However, there is a lack of control experiments, and additional experiments
are needed to determine whether the AncBE4 version works safely on zebrafish system. Here, I attached some suggestions to revise the study. 
  major
- What is the purpose of the P2A linked GFP co-expression?  According to the author's observation, it seems like that GFP does not work under the zebrafish experimental conditions. If authors can't show that GFP works in the BE4max and AncBE4max context, it seems to have no meaning to show.
-In Table 2 and main text (page 10, lane339-341), BE3 and BE4 control is missing. It would be much better to compare the genome editing activity between BE3 version and advanced one (BE4max or AncBE4max) in author's experimental condition. -In main text (page 8, lane 271-273), I think overall off-target analysis is lacking. According to recent reports (ref1,2), there are many CRISPR-Cas9 module dependent or independent off-target results. Even if it is not a whole-genome scale, authors must analyze off-targets which has similar sequence to on-target.
minor
-page2, lane 63-65; lane 68-69; lane 69-74 references should be attached.
- page4, lane159;  Details of somatic editing procedure is required.
  Ref1. Evaluation and minimization of Cas9-independent off-target DNA editing by Cytosine base editiors. Nature biotechnology
Ref2. Genome-wide target specificities of CRISPR RNA-guided programmable deaminases. Nature biotechnology

Author Response

Response to Reviewer 1 Comments

The authors applied a new version of the BE4max or AncBE4max base editor

to zebrafish to facilitate the creation of disease models. This is important because it extends the genome editor toolbox to make genome editing more precise in zebrafish system. However, there is a lack of control experiments, and additional experiments

are needed to determine whether the AncBE4 version works safely on zebrafish system. Here, I attached some suggestions to revise the study.

Major

Point 1:  What is the purpose of the P2A linked GFP co-expression? According to the author's observation, it seems like that GFP does not work under the zebrafish experimental conditions. If authors can't show that GFP works in the BE4max and AncBE4max context, it seems to have no meaning to show.

Response 1: The purpose of P2A linked GFP is for easy identification of embryos that are expressing the injected base editor mRNA. However, it is not as important if a skilled technician performs microinjections as is the case in our laboratory. We agree with the reviewer that since we did not observe any GFP in our injected embryos, showing data from P2A linked GFP versions of BE4max and AncBE4max does not add to the paper. Therefore, we have removed these data as follows:  

Changes to text (Lines 223-235): Deleted the text describing results of GFP plasmids and revised remaining text appropriately.

Changes to Figures: Removed Figure 3, and Figure S1 panel D (All representing data from GFP plasmids), Figure S1(panels A-D) is new Figure 3 (Data from non-GFP plasmids), figure legends were revised accordingly.

Changes to Tables: Removed data for GFP plasmids from Table 2

Point 2: In Table 2 and main text (page 10, lane339-341), BE3 and BE4 control is

missing. It would be much better to compare the genome editing activity between BE3 version and advanced one (BE4max or AncBE4max) in author's experimental condition.

Response 2: We agree with the reviewer that BE3 and BE4 data under our experimental conditions is missing. Since our aim was to evaluate the improved base editors, we chose the targets for which BE3 data was published although using different methods of evaluation. It would be difficult to perform these experiments in 10 days allowed for revision and we feel confident that even without these data, zebrafish researchers would appreciate our study showing that the new editors did not produce any indels. We have revised text in lines 339-341 (now lines 356-358) as follows “In conclusion, our study showed that BE4max and AncBE4max can be used to generate heritable point mutations in zebrafish without unintended indels.”

Point 3: In main text (page 8, lane 271-273), I think overall off-target analysis is lacking. According to recent reports (ref1,2), there are many CRISPR-Cas9 module dependent or independent off-target results. Even if it is not a whole-genome scale, authors must analyze off-targets which has similar sequence to on-target.

Ref1. Evaluation and minimization of Cas9-independent off-target DNA editing by Cytosine base editors. Nature biotechnology

Ref2. Genome-wide target specificities of CRISPR RNA-guided programmable deaminases. Nature biotechnology

Response 3: We understand the Reviewer’s concern about off-target base editing at other sites in the genome. One advantage of zebrafish model is that once a mutant line is established, out-crossing and genotype-phenotype correlations can be used to ensure that the observed phenotype is not due to other changes in the genome. However, off-target changes at a closely linked site can be inherited with the mutation, thus we routinely perform mismatch analysis of target sites during the design process. Both targets used in this study show sites in the genome with 4 or more mismatches, however none of the sites were on the same chromosome and thus not of concern. We have revised the text as follows to add an explanation on genome-wide off target effects:

Results: Lines 276-278: “We did not evaluate our fish for genome-wide off target effects, as we can use out-crossing and genotype-phenotype correlations to analyze the mutant fish.”

Discussion: Revised lines 328-333 to add three new references (#50,51,52) including the two mentioned by the reviewer about genome-wide off target edits.

Minor

Point 4: page2, lane 63-65; lane 68-69; lane 69-74 references should be attached.

Response 4: We thank the reviewer for pointing out the missing references for this paragraph. We have added appropriate references in this paragraph at the end of several sentences. Most of the papers were already cited in the next paragraph, however we have added two new references: Gaudelli et al., 2017 (#31) for adenine deaminases and Anazalone et al, 2020 (#29) for a base editing review.

Point 5:  page 4, lane159; Details of somatic editing procedure is required. 

Response 5: We thank the reviewer for bringing this to our attention. We have added the following sentence (Lines 161-163): “From each injection group, 24 - 48 embryos were collected individually in a 96-well PCR plate at 48 hpf and processed for DNA extraction, PCR and Sanger sequencing using the M13F primer as described above.”

Reviewer 2 Report

This appears to be a nicely done study on the utility of BE4max and AncBE4max for base editing in zebrafish.  These reagents are not new, but their use in zebrafish has not yet been reported.  As hoped, these base editing constructs can deliver zebrafish lines with desired point mutations without the introduction of unwanted indels.  Further optimization is clearly needed however, as incorrect conversion of nucleotides and bystander edits were both observed.  However, it seems like these reagents are robust enough that determined investigators can indeed employ them for base editing the zebrafish genome.

The manuscript is straight forward, easy to read, and describes a very logical work flow that could serve as a template for other investigators interested in base editing work.

One caveat to my review, I do not have ANY experience in zebrafish work.  For example, I was unaware that males must be present for ovulation to occur, and thus not possible to screen unfertilized embryos.

Minor comments and suggestions:

Line 61 ... several versions with homology arms of different lengths need to ...

Line 66 ... using a nickase version ...

Line 123:  ... in red in front of the PAM in ....

Line 131: ... Plasmid DNA was purified from ???? using a miniprep kit Qiagen).

Line 182 ... to target the twist locus ....

Line 183:  .... to target the ntl locus ....

Line 330: AncBE4max is codon optmized for what species, human?

How different is codon usage in mammals versus zebrafish?  Would a zebrafish optimized construct really be expected to make a difference relative to human or mouse optimized?  What were the zABE7.10  tools compared to.

Author Response

Response to Reviewer 2 Comments

This appears to be a nicely done study on the utility of BE4max and AncBE4max for base editing in zebrafish. These reagents are not new, but their use in zebrafish has not yet been reported. As hoped, these base editing constructs can deliver zebrafish lines with desired point mutations without the introduction of unwanted indels. Further optimization is clearly needed however, as incorrect conversion of nucleotides and bystander edits were both observed. However, it seems like these reagents are robust enough that determined investigators can indeed employ them for base editing the zebrafish genome.

The manuscript is straight forward, easy to read, and describes a very logical work flow that could serve as a template for other investigators interested in base editing work.

One caveat to my review, I do not have ANY experience in zebrafish work. For example, I was unaware that males must be present for ovulation to occur, and thus not possible to screen unfertilized embryos.

Minor comments and suggestions:

Point 1:  Line 61 ... several versions with homology arms of different lengths need to ...

Response 1: We have made this change as suggested.

Point 2: Line 66... using a nickase version ...

Response 2: We have made this change as suggested.

Point 3: Line 123: ... in red in front of the PAM in ....

Response 3: We used red and blue fonts in the Table to mark PAM and target C’s. We have changed the word “font” to “color” to make it clear.

Point 4: Line 131: ... Plasmid DNA was purified from ???? using a miniprep kit Qiagen).

Response 4: We have revised this sentence as “Plasmid DNA was purified from an overnight culture of TOP10 cells (ThermoFisher Scientific) using a miniprep kit (Qiagen).”

Point 5:  Line 182 ... to target the twist locus ....

Line 183: .... to target the ntl locus ....

Response 5: We have added “the” in both lines as suggested by the Reviewer (Now lines 185, 186)

Point 6:  Line 330: AncBE4max is codon optimized for what species, human?

Response 6: Line 330 (now line 343) describes ABE7.10 and ABEmax 7.10 versus their zebrafish versions. The original constructs ABE7.10 (Gaudelli et al., 2017) and ABEmax 7.10 (Koblan et al., 2018) were optimized for use in mammalian cells. They differed by the use of SV40 NLS (ABE7.10) versus codon-optimized bis bpNLS (ABEmax7.10). Qin and colleagues (2018) replaced both adenosine deaminase (TadA and TadA*) and nCas9 of original constructs with zebrafish codon-optimized counterparts (zTadA, zTadA*, znCas9). We have revised text to add this information in lines 343-346. AncBE4max is described in lines 90-91 and we have revised it to make it clear that the rAPOBEC-1 was replaced with APOBEC optimized by ancestral reconstruction in AncBE4max.

Point 7:  How different is codon usage in mammals versus zebrafish? Would a zebrafish optimized construct really be expected to make a difference relative to human or mouse optimized? What were the zABE7.10 tools compared to.

Response 7: Codon optimization is recommended to enhance expression of heterologous genes (See Zhou et al., Codon usage is an important determinant of gene expression levels largely through its effects on transcription, PNAS Sept 26, 2016: E6117-E6125). Thus, various DNA synthesis companies (e.g. IDT, GenScript) have developed codon optimization tools that can be used to synthesize recombinant DNA specific for expression into a particular species. Previous studies in zebrafish have demonstrated a significant improvement in genome editing efficiencies using zebrafish optimized Cas9 (Jao et al PNAS 2013). Qin and colleagues compared zABE7.10 and zABEmax7.10 with their original counterpart, pCMV-ABE7.10. We have revised the text and added the Jao et al reference (#53) on codon optimization in zebrafish to lines 341-343.

Reviewer 3 Report

The manuscript entitled “BE4max and AncBE4max are efficient in germline conversion of C:G to T:A base pairs in zebrafish” reported the use of BE4max and AncBE4max in editing zebrafish. Both somatic and germline editing rates were evaluated, and AncBE4 showed a higher efficiency than BE4max according to their results. Overall, this draft is properly designed and well organized. I have only a few minor concerns:

  1. Line 265, I don’t believe PCR can induce a error rate of 4%. Repeating the PCR and/or sequencing can simply rule out the possibility.
  2. Line 196. By saying ”repeating injections”, do you mean injecting the same embryo for multiple times?
  3. Since you used previously reported sgRNAs in BE3-associated studies, I think it is a good idea to compare the efficiency in this study with those reported using BE3.
  4. Overall limitations on BE systems can be further discussed.

Author Response

Response to Reviewer 3 Comments

The manuscript entitled “BE4max and AncBE4max are efficient in germline conversion of C:G to T:A base pairs in zebrafish” reported the use of BE4max and AncBE4max in editing zebrafish. Both somatic and germline editing rates were evaluated, and AncBE4 showed a higher efficiency than BE4max according to their results. Overall, this draft is properly designed and well organized. I have only a few minor concerns:

Point 1: Line 265, I don’t believe PCR can induce a error rate of 4%. Repeating the PCR and/or sequencing can simply rule out the possibility

Response 1: We agree with the reviewer that PCR does not induce errors at such high rate. Our writing was not clear, as these 3 changes are all different, so each one could be an independent PCR error. We can’t go back to the same samples to repeat PCR. Therefore, we have revised the text by deleting the following sentence (lines 271-273): “Since the expected nucleotides were not edited in these embryos, and their frequency was low, we speculate that these might be PCR induced errors rather than a result of off-target activity of AncBE4max.”

Point 2: Line 196. By saying ”repeating injections”, do you mean injecting the same embryo for multiple times?

Response 3: Injections are done in 1-cell stage embryos, so same embryos can’t be injected again. To make it clear we have revised the sentence as follows (Lines 199-200): “This was done by injecting ~100 additional 1-cell stage embryos and growing them to adulthood.”

Point 3: Since you used previously reported sgRNAs in BE3-associated studies, I think it is a good idea to compare the efficiency in this study with those reported using BE3.

Response 3: We thank the reviewer for this suggestion. Due to differences in methods of evaluating somatic base editing efficiency, we could not directly compare their data with ours. Therefore, we compared the efficiency of identifying germline transmitting founders in both studies and added a sentence to Discussion (Lines 307-309) as follows: “The percentage of germline transmitting founders in our study (25-28%) is comparable to that reported with BE3-zCas9 (28-37%) at the same two loci (citation).”

Point 4:  Overall limitations on BE systems can be further discussed.

Response 4: We have addressed two major limitations of BE systems: precision of editing and efficiency of editing in great detail in Discussion. We have now revised it to add about genome-wide off target effects (Lines 328-335), and overcoming PAM site restrictions by swapping nCas9 with its variant versions or Cpf1 (Lines 348-350).

Round 2

Reviewer 1 Report

The authors compensated for the shortcomings of this paper by revising the text.